# Technical Note: A validated correction method to quantify organic and inorganic carbon in soils using Rock-Eval® thermal analysis

Marija Stojanova[1], Pierre Arbelet[2], François Baudin[3], Nicolas Bouton[4], Giovanni Caria[5, 6], Lorenza Pacini[1,2], Nicolas Proix[5], Edouard Quibel[2], Achille Thin[2], and Pierre Barré[1]

[1]Laboratoire de Géologie, École Normale Supérieure, CNRS, Université PSL, IPSL, Paris, France
[2]Greenback (commercial name: Genesis), Paris, France
[3]UMR ISTeP 7193, Sorbonne Université, CNRS, Paris, France
[4]Vinci Technologies, 27B rue du Port F-92000 Nanterre, France
[5]INRAE, Laboratoire d'Analyses Des Sols, Saint-Laurent Blangy, France
[6]Univ. Lille, CNRS, UMR 8516, LASIRE, Equipe Physico-Chimie De l'Environnement, Lille, France

**Correspondence:** Marija Stojanova (stojanova@ens.fr) and Pierre Barré (barre@ens.fr)

**Abstract.** Soils contain large amounts of carbon stored as organic carbon and carbonates. These carbon pools can contribute to climate regulation, and are of primary importance in ensuring proper soil functioning. However, their accurate quantification remains a complex task. Rock-Eval® thermal analysis has emerged as an alternative to classic dry combustion and wet methods, due to its ability to simultaneously provide organic and inorganic carbon measurements on the same subsample. However, it has been observed that Rock-Eval® systematically underestimates the soil organic carbon (SOC), while overestimating the soil inorganic carbon (SIC). In this technical note, we propose a validated correction of both SOC and SIC based on a machine-learning model and using a diverse dataset of 240 soil samples. We show that the proposed correction significantly increases the accuracy of the Rock-Eval® method when compared to reference SOC and SIC values when applied to the dataset used for training and testing, and that it can be successfully applied to data originating from different Rock-Eval® machines, without changing the routine analytical protocol. The transferability of the model allows for its future implementation in the Geoworks software so that Rock-Eval® machines can routinely provide accurate SIC and SOC measurements.

## 1 Introduction

Soils contain large amounts of carbon stored as organic carbon and carbonates (Batjes, 1996; Zamanian et al., 2018). These carbon pools are dynamic and can contribute to climate regulation (Chenu et al., 2019; Zamanian et al., 2021). Moreover, organic carbon is also of primary importance in ensuring proper soil functioning (Hoffland et al., 2020). Global stocks of SIC and SOC are of comparable size ($\sim$2500 Pg) when the soil is considered down to a depth of 2 meters (Zamanian et al., 2021). Not all soils contain SIC, but the presence of carbonates is frequent, particularly in arid or semi-arid regions (Zamanian et al., 2018; Pfeiffer et al., 2023). The presence of SIC influences soil pH and therefore nutrient availability (e.g., Mkhonza et al. (2020)). This explains the usual agricultural practice of liming to reduce soil acidity. In addition, recent studies have shown that the amount of SIC can vary significantly over short time scales ($\sim$10 years) due to soil acidification resulting from certain farming practices such as nitrogen fertilization, irrigation, etc. (Zamanian et al., 2021; de Soto et al., 2017), suggesting that it

may be worthwhile to assess the potential role of carbonates as a source or sink of C in a context of climate change. Moreover, land-based enhanced rock weathering is increasingly discussed as a strategy to contribute to atmospheric $CO_2$ removal. It consists of the spreading of silicate powder whose weathering in soils leads to atmospheric $CO_2$ sequestration in carbonates (e.g., Kelland et al. (2020)). Measuring the soil carbonates produced can provide information on the effectiveness of adding silicate minerals to sequester atmospheric $CO_2$ Haque et al. (2020). As a result, the accurate quantification of soil organic and inorganic carbon (respectively SOC and SIC) content is recognized as an increasingly important, yet not straightforward task, especially in carbonated soils. There exists a wide range of methods for quantifying SOC and SIC, the two main approaches are dry and wet combustion methods, and they both come with their advantages and drawbacks.

Dry combustion methods involving CHN (Carbon, Hydrogen, Nitrogen) elemental analyzers are increasingly used to measure total carbon. If the soil sample is not carbonated, the total carbon therefore represents only SOC. In carbonated soils however, the amount of SIC is quantified using a separate analysis on a second subsample and the quantity of SOC is then determined as the difference between total carbon and SIC. SIC is generally measured using the amount of $CO_2$ produced by a known quantity of soil after acidification with HCl (Allison and Moodie, 1965). Alternatively, SOC can be directly quantified using CHN if carbonates have been previously removed using acid fumigation (Harris et al., 2001). The standard ISO 10694 method (ISO, 1995, 1999) describes how organic, inorganic, and total carbon can be measured in soil samples using dry methods. The main disadvantages of dry methods are that (1) SIC and SOC quantities are not measured on the same subsamples, and (2) both SOC and SIC measurements can be inaccurate if the decarbonatation is incomplete.

Wet methods are also largely used, especially when CHN analyzers are not available. The most commonly used wet oxidation method is the Walkley-Black method that allows measuring organic carbon directly in carbonated and non-carbonated soils (Walkley and Black, 1934). However, the Walkley-Black method has an average SOC yield of 77 % and therefore the results have to be corrected to account for not recovered SOC (Food and Agriculture Organization of the United Nations, 2020). The correction factor can be significantly different from one soil to another (Neal and Younglove, 1993), representing a limitation for the accuracy of wet oxidation methods (ISO 14235:1998 Soil Quality, 1998).

More recently, thermal analysis methods have been proposed as an interesting alternative to quantifying both SOC and SIC on a single subsample (Disnar et al., 2003; Vuong et al., 2016; Apesteguia et al., 2018). Among thermal methods, Rock-Eval® (RE) thermal analysis has received a particular attention in the recent years (Koorneef et al., 2023; Hazera et al., 2023), suggesting that if suitable corrections are applied, the Rock-Eval® method could provide accurate quantification of both SIC and SOC. Moreover, a recent study has shown that the results provided by the Rock-Eval method are highly repeatable and reproducible, with relative errors of the order of 2 to 4 % for the measurements of organic and inorganic C contents (Pacini et al., 2023). The aim of this technical note is to provide appropriately validated corrections to enable quantification of both SIC and SOC by Rock-Eval® and to test the transferability of corrections between Rock-Eval® machines.

## 2 Materials and methods

### 2.1 Soil samples

The study mobilized 240 soil samples taken from the surface layer (0-30 cm) of French agricultural soils. A stratification procedure was used to select the samples and ensure that each combination of texture class, SIC quantity, and SOC quantity was represented with at least 7 samples for low-carbonated soils, and 14 samples for carbonated soils. As a result, the selected samples cover a wide range of particle-size distribution, and SIC and SOC content, as shown in Fig. 1. SIC and SOC quantification was conducted at the Soil Analyses Laboratory (LAS) of Saint-Laurent Blangy, accredited by Cofrac (French accreditation committee). LAS is a public laboratory operated by INRAE (French National Research Institute for Agriculture, Food, and Environment) that analyzes samples for the French and European research community. At LAS, the total carbon is determined using a CHN analyzer (Thermofisher Flash 2000) and inorganic carbon is quantified using a Bernard calcimeter. In carbonated samples, organic carbon is then determined as the difference between total carbon and inorganic carbon. SIC and SOC measured at the LAS can be considered as highly accurate and are therefore suitable to constitute reference measurements. Their associated uncertainties are calculated as:

$$\text{Uncertainty SOC} = 0.02 \times ([\text{SOC}] + 0.12 \times [\text{CaCO}_3]) + 0.49 + 0.12 \times (0.016 \times [\text{CaCO}_3] + 0.63), \text{in g kg}^{-1}, \tag{1}$$

and

$$\text{Uncertainty SIC} = 0.12 \times (0.016 \times [\text{CaCO}_3] + 0.63), \text{in g kg}^{-1}. \tag{2}$$

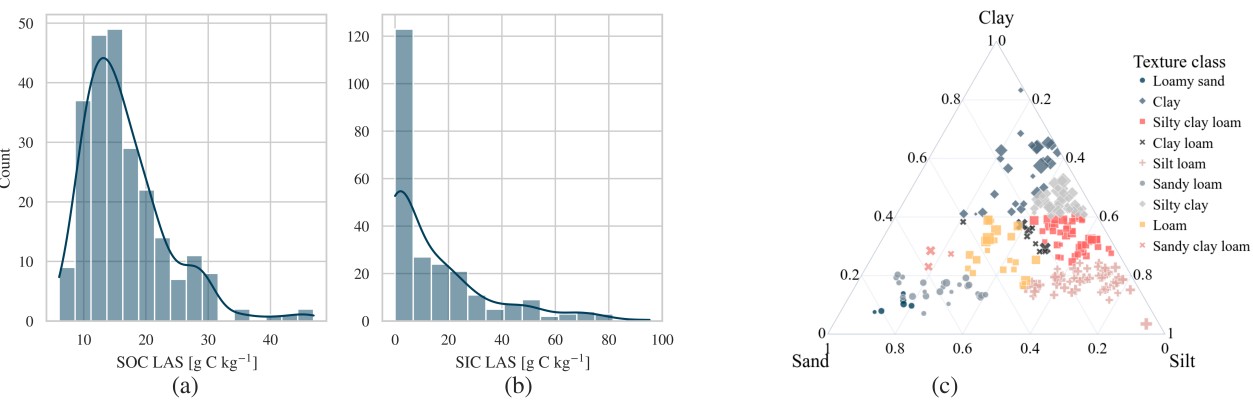

**Figure 1.** Distributions of SOC (a), SIC (b), and particle-size as a proportion of sand, silt, and clay (c), in the dataset of 240 samples, as measured by the LAS Saint-Laurent Blangy. SIC range is 0.5 to 96.8 g C kg$^{-1}$, and SOC range is 5.37 to 44.64 g C kg$^{-1}$.

### 2.1.1 Rock-Eval® measurements

Each of the 240 soil samples were ground using a Fritch pulverisette 6 tungsten carbide planetary mill and divided in three subsamples. One subsample was sent to LAS Saint-Laurent Blangy for SIC and SOC determination. The two other subsamples were analyzed using Rock-Eval® at ISTeP (Earth Science Institute, CNRS, Sorbonne University, Paris, France), and at Vinci Technologies (Nanterre, France). At ISTeP, samples were analyzed using a Rock-Eval® 6 Turbo (RE6Turbo), whereas at Vinci Technologies samples were analyzed using an RE6 Standard instrument and a Rock-Eval® 7 (RE7) instrument. The RE7 is the

new version of Rock-Eval® machines and we therefore considered it useful to check whether a correction designed on RE6 can be transferred on the new RE7 version.

The Rock-Eval® thermal analysis consists of a pyrolysis step, followed by an oxidation step. During pyrolysis, the sample is gradually heated in an inert environment ($N_2$) and evolved hydrocarbons, $CO_2$, and CO effluents are continuously monitored with time. Then, the sample is cooled down before the start of the oxidation stage. During oxidation, the sample is gradually

heated in purified laboratory air and the quantity of evolved $CO_2$ and CO is continuously monitored. The used analytical setup was the "SOIL" routine. The pyrolysis step started with a 3 min isotherm at 200° C followed by a heating ramp of 30° C $min^{-1}$ from 200° C to 650° C (Disnar et al., 2003). The heating routine for the oxidation step started with a 1 min isotherm at 300 ° C followed by a heating ramp of 20° C $min^{-1}$ until 850° C and terminated with a 5 min isotherm at 850° C (Baudin et al. (2015), adapted from Behar et al. (2001)). The correct determination of SIC using the original settings of the widely used "SOIL"

routine, even for carbonate-rich samples (see for instance Delahaie et al. (2023)), suggests that there is no need to modify the Rock-Eval acquisition parameters for carbonated soils as suggested by Hazera et al. (2023). Pacini et al. (2023) details the RE6 thermal analysis, as well as the usage of the "SOIL" routine in Geoworks.

All samples (ca. 60 mg) on all machines were analyzed using this routine. A reference sample (IFPEN_160000) is measured regularly in Rock-Eval® batch analyses for quality control. The target values are $3.28 \pm 0.06$ wt% and $3.26 \pm 0.07$ wt% for

SOC and SIC, respectively. The machine is recalibrated when SOC and SIC values are outside this narrow range meaning that the acceptable measurement error is ca. 2%. The thermograms were integrated to provide RE parameters and indices using the Geoworks v1.7 software. Notably, TOCre6 and MinC (corresponding to Rock-Eval® SOC and SIC measurements) were provided by Geoworks. The thermograms showed that the carbonate soil samples contained neither dolomite nor siderite.

### 2.2 Design of the correction methods

It has been long known that TOCre6 and MinC tend to slightly (ca. 10%) underestimate SOC and overestimate SIC, respectively (Disnar et al., 2003). The data we collected from the LAS Saint-Laurent Blangy and the Rock-Eval® machines (ISTeP and Vinci Technologies) also respects this general tendency, as shown in Fig. 2. The goal of this study is to design and validate a correction method for soils that would align the TOCre6 and MinC values measured by Rock-Eval® as closely as possible with reference SOC and SIC measurements provided by the LAS. To do so, we considered several correction methods, each one

implementing a different Machine Learning (ML) model. We began by learning a correction method using the data obtained with the RE6 Turbo from ISTeP and pairing it with the soil organic and inorganic carbon data from LAS Saint-Laurent Blangy.

We then tested the generalizability of the correction by applying it to the data obtained using other machines, i.e., the RE6 and RE7 data from Vinci Technologies. In the interest of brevity and legibility, certain modeling choices and techniques are omitted from the main text and are instead included in our Supplementary materials in the form of a Jupyter notebook (Kluyver et al., 2016) using open-source code Python 3 code (Van Rossum and Drake, 2009).

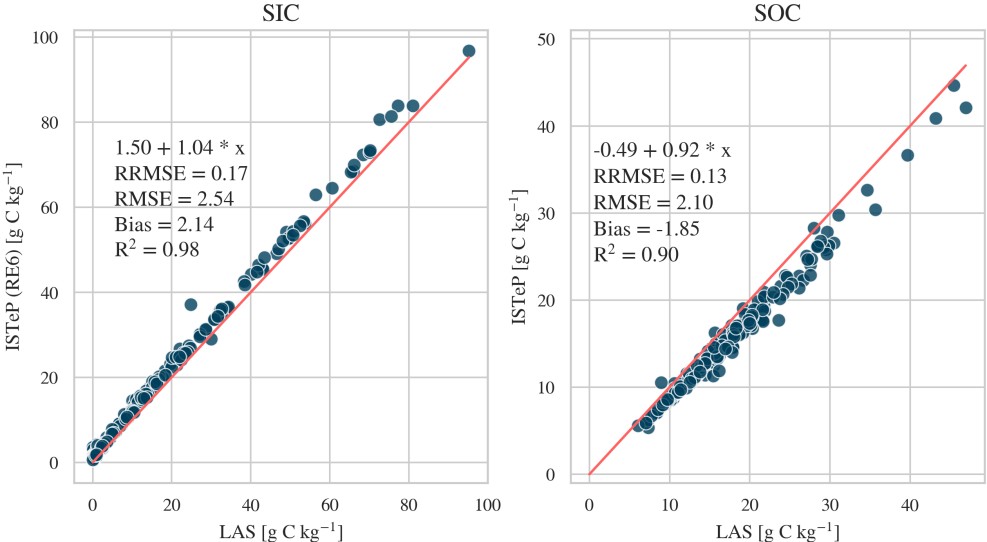

**Figure 2.** Comparison of the SIC and SOC measurements before correction from the RE6 analysis by ISTeP and reference measurements by LAS Saint-Laurent Blangy. The red line represents the 1:1 diagonal.

After having tested several different approaches, the corrective model we propose here is a Support Vector Machine (SVM) (Cortes and Vapnik, 1995) regression as implemented in the scikit-learn Python library (Pedregosa et al., 2011). SVMs are a collection of supervised ML models that are well-suited to our usage as they adapt to both linear and non-linear data. SVMs using the RBF (Radial Basis Function) kernel offer two different regularization hyper-parameters, which allows us to avoid overfitting the training data. Other models we have tested are a regular least-squares Linear Regression (LR), a constrained Ridge Regression (Ridge), and a Random Forest (RF) model. The LR model is an ordinary least squares linear regression that minimizes the residual sum of the differences between the observed and the predicted data. The Ridge regression is a special case of LR where regularization is added to the coefficients using the l2-norm, so as to avoid overfitting the training data. Lastly, the RF is an ensemble modeling method that uses averaging of a collection of randomized decision trees, i.e., random subset of features and of samples, to predict the target variable. Like SVMs, RFs are capable of capturing linear and non-linear relationships in the data. After correcting the SIC and SOC, it can happen that the corrected value is a small negative number close to zero, especially for low-carbonated soil samples. In this case, we simply set these negative SIC and SOC values to zero.

The dataset contains all the RE parameters (such as Hydrogen Index (HI), Oxygen Index (OI), Pyrolyzable Carbon (PC), Temperature stability parameters etc., see Pacini et al. (2023) and Baudin (2024) for an exhaustive list) for the 240 soil samples including the TOCre6 (namely $SOC_{RE6}$) and MinC (namely $SIC_{RE6}$), as well as SOC and SIC provided by the LAS.

We develop two separate models, for predicting the needed correction of the SOC and the SIC. Our target variable is the difference between the SOC (respectively SIC) measured by the LAS and the RE6 analysis:

$$y_{SOC} = SOC_{CHN} - SOC_{RE6} \; ; \tag{3}$$

$$y_{SIC} = SIC_{CHN} - SIC_{RE6} \; . \tag{4}$$

All the models are crossed-validated during the training phase using two-thirds of the available data, and then tested on the remaining one-third of samples. After a correlation analysis between the target variables and the RE6 parameters, as well as running several multivariable models, we concluded that the Rock-Eval® SOC (respectively SIC) alone is sufficient to accurately estimate the needed correction. This procedure begins by calculating the Spearman correlation, where it was evident that the $SOC_{RE6}$ and $SIC_{RE6}$ are the most highly-correlated with the target variables. Next, we trained the four ML models mentioned above (SVM, LR, Ridge, RF) using all or subsets of the RE parameters that were highly correlated with the target. We used two different thresholds of 0.3 and 0.5 in Spearman correlation for detecting the highly-correlated parameters, allowing us to have a smaller or larger set of parameters. We also removed any parameters that were highly-correlated among each other, this time using a threshold of 0.9 in Spearman correlation. The results of all these tests were overwhelmingly in favor of using only the $SIC_{RE6}$ and $SOC_{RE6}$ in the correction model.s Therefore, in the remainder of this note, all the proposed corrections are single-feature models. Finally, the corrected SIC and SOC of RE can be calculated as:

$$SOC_{corrected} = SOC_{RE6} + \widehat{y}_{SOC} \; , \tag{5}$$

and

$$SIC_{corrected} = SIC_{RE6} + \widehat{y}_{SIC} \; , \tag{6}$$

where $\widehat{y}_{SOC}$ and $\widehat{y}_{SIC}$ are the predicted corrections of the ML model.

For the sake of comparison, we also implemented the corrective approach described by Hazera et al. (2023) derived from a seminal study by Disnar et al. (2003). In this study, two different corrections are proposed for SOC in calcareous samples depending on the state of degradation of organic matter. The proposed correction for samples enriched in fresh organic matter is thereafter named $Disnar_{BRS}$ for Biopolymer-Rich Samples, whereas the correction for samples containing more processed organic matter is thereafter named $Disnar_{soil}$. As the degradation state of soil organic matter sensu Disnar et al. (2003) is

not easy to determine, we decided to consider both corrections in our study for carbonated and non-carbonated samples. The correction for carbonated biopolymer-rich samples ($Disnar_{BRS}$) is:

$$SOC_{corrected} = 1.17 \times SOC_{RE6} \,, \tag{7}$$

and

$$SIC_{corrected} = SIC_{RE6} - 0.092 \times SOC_{RE6} \,. \tag{8}$$

While for non-carbonated samples they propose the correction $Disnar_{soil}$:

$$SOC_{corrected} = (SOC_{RE6} + SIC_{RE6}) \times 1.068 \,, \tag{9}$$

and

$$SIC_{corrected} = 0 \,. \tag{10}$$

For non BRS soils, they propose the following correction, named hereafter $Disnar_{soil}$, for carbonated soils:

$$SOC_{corrected} = SOC_{RE6} + 1.092 \times SIC_{RE6} \,, \tag{11}$$

and

$$SIC_{corrected} = SIC_{RE6} - 0.092 \times SIC_{RE6} \,. \tag{12}$$

and for non-carbonated soils:

$$SOC_{corrected} = SOC_{RE6} + SIC_{RE6} \,, \tag{13}$$

and

$$SIC_{corrected} = 0 \,. \tag{14}$$

As the Disnar et al. (2003) model divides the soil samples into carbonated and non-carbonated, we also divide our data using a similar threshold of 2 g C $kg^{-1}$ of SIC. This approach results in three different datasets: the non-carbonated (80 samples), carbonated (160 samples), and all data (240 samples). In the accompanying Notebook, we have detailed a second threshold used to distinguish carbonated and non-carbonated samples based on the contribution to the $S5$ signal in the RE6 analysis. Due to the similarity of the two approaches, we have only presented the former in the following of this publication.

### 2.3 Metrics to evaluate the correction methods

We use several different metrics to quantify the quality of the provided correction. The $R^2$ coefficient of determination is the proportion of the variable in the dependent variable that can be explained by the independent variable. Its upper bound and best

score is 1, while its lower bound is minus infinite. An $R^2$ score of 0 means that none of the dependent variable's variability is explained and the model only predicts the mean value. It is calculated as:

$$R^2 = 1 - \frac{\sum(y_i - \widehat{y}_i)^2}{\sum(y_i - \overline{y})^2} \; , \tag{15}$$

where $\widehat{y}_i$ is the predicted value of the i-th sample and $\widehat{y}$ is the average value over all samples. The Root Mean Squared Error (RMSE) and Relative RMSE (RRMSE) are calculated as follows:

$$\text{RMSE} = \sqrt{\frac{\sum(y_i - \widehat{y}_i)^2}{N}} \; ; \tag{16}$$

$$\text{RRMSE} = \frac{\text{RMSE}}{\overline{y}} \; , \tag{17}$$

where N is the number of samples. We also calculate the bias as the difference between the mean of the measured values and the mean of the predicted values. The bias allows us to know if the model tends to systematically over- or under-estimate the target variable.

## 3    Results and discussion

### 3.1    Compared performances of the corrective methods

Having introduced several models for both the SIC and the SOC, we proceed with assessing their corrective performance on the test dataset. The metric we are mainly interested in is the RMSE, and to a lesser extent the Bias, the $R^2$ coefficient of determination, and the ordinary least squares linear equation describing the relationship between the reference CHN values provided by the LAS and the (corrected) RE values.

Figure 3 shows the RMSE for the original data and of the corrected data after using one of the four different tested ML models and the correction factors provided by Disnar et al. (2003). Concerning the SIC correction, the overall best performance is achieved by the SVM model, as it is the only one to significantly reduce the error in all three datasets (carbonated, non-carbonated, and all samples). With the exception of the RF model for non-carbonated data, all the models we propose offer a considerable decrease in RMSE. The RF model, due to its intrinsic complexity and bootstrapping, is probably more impacted than other ML models by the smaller dataset size, as only the 80 non-carbonated samples are considered, and one-third of those are set aside for testing. Regarding the SOC correction, the models we propose strongly increase the similarity between LAS and Rock-Eval® ISTeP SOC values. They have virtually equal performance, resulting in a two- to three-fold decrease in RMSE. A more detailed comparison of the behavior of the models as a function of the input data is available in the Supplementary material.

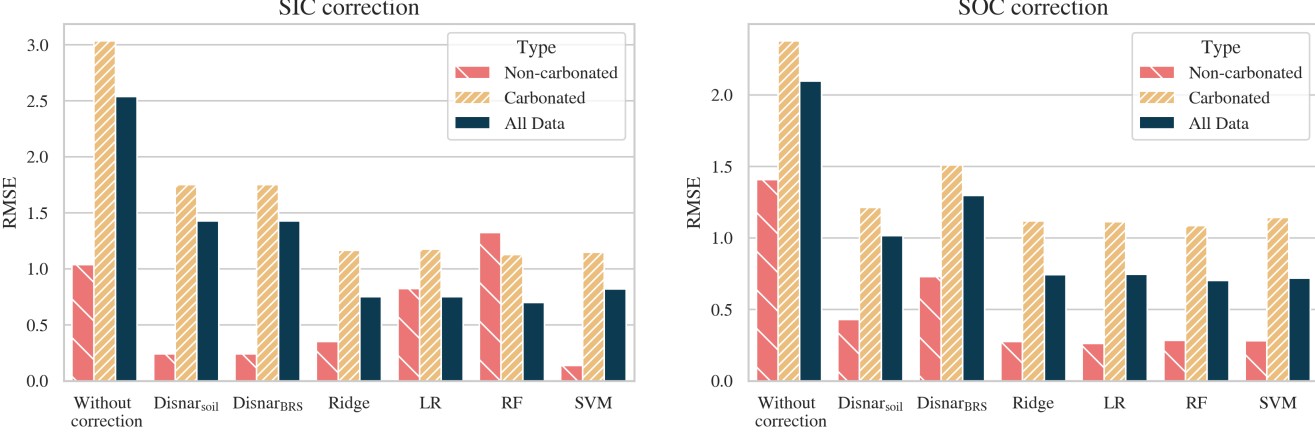

**Figure 3.** Correction accuracy comparison of the four tested ML models, the Disnar et al. (2003) corrections Disnar$_{BRS}$ and Disnar$_{soil}$, and the original data before correction.

The correction factors proposed by Disnar et al. (2003) significantly decreased the RMSE for SIC and SOC for all datasets. However, Fig. 3 clearly shows that this correction is not optimal as the RMSE is on average about twice higher compared to the ML-based corrections. This is not very surprising as, contrary to the other methods, the correction proposed in Disnar et al. (2003) is not derived from an optimized model trained and tested on separate data. Even though the four ML-based corrections render highly similar performance, the overall best-performing model is the SVM. A paired T-test comparison between the Disnar and SVM models is available in the Notebook that shows the two corrections provide significantly different results. The results of the SVM correction are shown in Fig. 4. The RMSE is divided by a factor of more than 3 for the SIC and more than 2 for the SOC with respect to the raw data before correction. For both the SIC and the SOC, the equation describing the relationship between corrected RE6 and LAS data is almost identical to the 1:1 ratio, showing that our proposed correction closes the gap between the two methods.

### 3.2 Transferability of the corrective method to other Rock-Eval® machines

In order to test the transferability of our results to other Rock-Eval® machines, we apply our SVM model learned on ISTeP's RE6 Turbo machine to the data obtained using Vinci Technologies' RE6 and RE7 machines. The reference SIC and SOC data stays the same, i.e., provided by the LAS Saint-Laurent Blangy. Figure 5 shows the comparison for the RE6 (a and b) and RE7 data (c and d), respectively. The correction quality is comparable to that of the ISTeP data with, not surprisingly, only slightly higher RMSEs. The overall bias is not only of the same order but sometimes lower, in part due to the larger sample size as the original correction metrics are calculated on only one third of the dataset used as testing data. The possibility to transfer a correction model for SIC and SOC learned on one Rock-Eval® machine to another was somehow expected, as Pacini et al. (2023) observed that TOCre6 and MinC are very reproducible across different RE6 instruments and for RE6 and RE7 instruments.

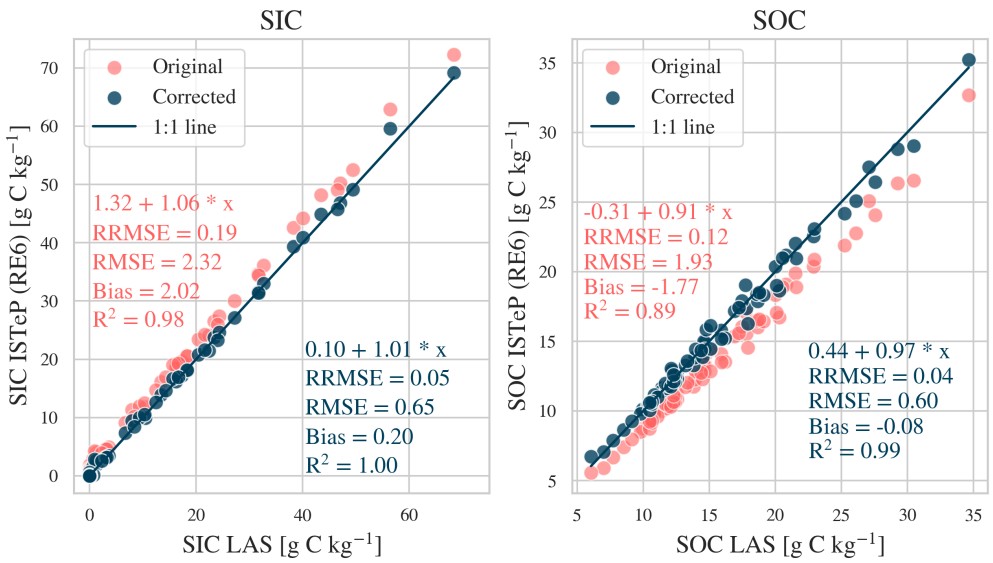

**Figure 4.** Precision of the proposed SVM correction on the test dataset.

## 4    Conclusions

Accurately estimating the quantity of soil organic carbon has important implications for the monitoring of soil health, as well as climate regulation techniques and policies. The classic dry combustion and wet methods, though standardized decades ago, come with a series of drawbacks when considering carbonated soils. Previous studies suggested that, on top of providing information on soil organic matter biogeochemical stability (Barré et al., 2016), thermal analyses could be an accurate means of determining SOC and SIC quantities in carbonated soil with lower experimental uncertainty. Our work confirms this hypothesis

and proposes the first validated correction method to accurately determine the quantities of SOC and SIC from a Rock-Eval® thermal analysis. This correction method, based on SVM machine-learning, can be transferred to different RE6 and RE7 instruments and will be implemented in the Geoworks software so that Rock-Eval® machines can routinely provide accurate SIC and SOC measurements, at least in the SOC and SIC value ranges investigated in this study (up to ca. 5% and ca. 8% for SOC and SIC respectively).

*Code and data availability.* The Jupyter notebook implementing the code and the data accompanying this technical note are both made available  (Stojanova et al., 2024).

        .

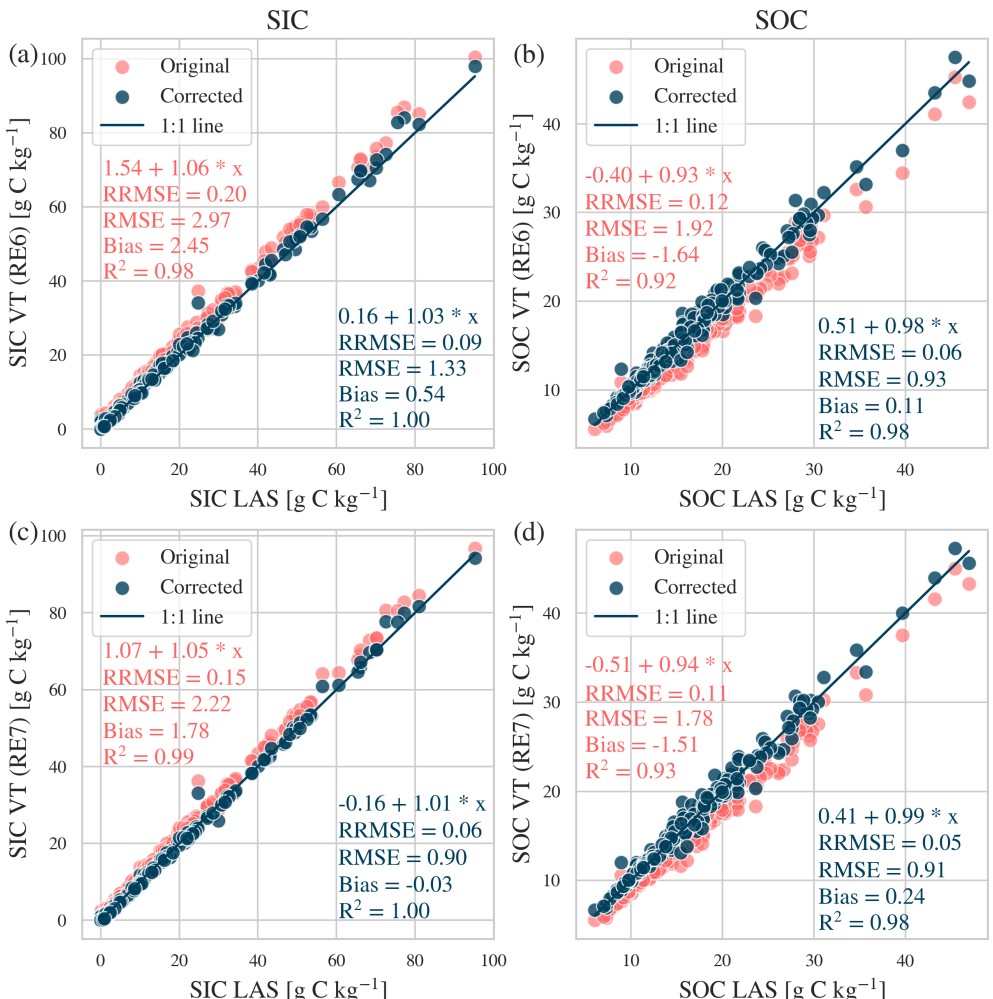

**Figure 5.** Results of applying the proposed SVM correction to data obtained using Vinci Technologies' RE6 and RE7 machines.

*Author contributions.* MS, PA, FB, NB, LP, and PB designed the study, MS conducted the data treatment in collaboration with PA, LP, and AT. PA and EQ provided the samples, GC and NP realized SIC and SOC reference measurements, FB and NB provided the Rock-Eval results, MS and PB wrote the first version of the draft which was corrected by all co-authors.

*Competing interests.* This work was funded by Vinci Technologies, which develops and markets the Rock-Eval machine.

*Acknowledgements.* The authors thank Florence Savignac (ISTeP) and Theirry Devillaz (Vinci Technologies) who carried out the Rock-Eval® analyses.

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
