# Peer review of "Technical Note: A validated correction method to quantify organic and inorganic carbon in soils using Rock-Eval® thermal analysis"

_EGUsphere, 2024_

## Author Response (AR1)

**Reviewer 1**

**General comments**

This technical note presents a correction method to determine soil organic content and soil inorganic content in single analysis. Using Rock-Eval thermal analysis offers some benefits over dry combustion method with elemental analyzers, where pretreatments or calculation are needed to get the two values of organic and inorganic carbon contents. Rock Eval eliminates the chances of calculation error, and experimental error associated with the elemental analysis method. The main conclusion by the authors is that Rock-Eval analysis can accurately determine SOC and SIC contents thanks to corrections based on a machine learning model. This result is highly promising to facilitate C studies in calcareous soils, however I have some reservations regarding the description, the application and the validation of these corrections. How did you definy the term "validated" ? Please explicit your criteria, the domain of validity and the correction itself? I hope that the authors can amend their manuscript to explicit the corrections, enhance the quality of the methodological work and discuss more their results in order to impede the paper to be taken as an advertisement for Rock-Eval® equipment with the Geoworks software. Please find some specific comments and questions below.

*Answer : We thank the reviewer for their careful reading of the draft and constructive comments.*

**Specific comments**

L.8 I did not understand the initial data set? What does it mean?

*Answer : The « initial data set » corresponds to the data obtained by analyzing the samples on the RE6 Turbo machine at Sorbonne University. This is the data set that was split into a learning and a validation set. The other data set refers to data obtained using RE6 and RE7 machines at Vinci Technologies. We recognize that this sentence is confusing and is modified as follows in the revised manuscript in line 8.*

Please add few lines on SIC, their distribution, the role of SIC in the soil properties, the eventual management of SIC (limestone inputs, Enhanced Rock Weathering…). Why is it important to consider the SIC contents as the SOC contents.

*Answer : We can indeed provide more information on the interest of quantifying SIC. We have added the following paragraph to the revised draft starting line 15:*

*Global stocks of SIC and SOC are of comparable size (~2500 Pg) when the soil is considered down to a depth of 2m (Zamanian et al., 2021). Not all soils contain SIC, but the presence of carbonates is frequent, particularly in arid or semi-arid soils (Zamanian et al., 2016; Pfeiffer et al., 2023). The presence of SIC influences soil pH and therefore nutrient availability (e.g., Mkhonza et al., 2020). This explains the usual agricultural practice of liming to reduce soil acidity.  In addition, recent studies have shown that the amount of SIC can vary significantly over short time scales (~10 years) due to soil acidification resulting from certain farming practices (nitrogen fertilization, irrigation, etc.) (Zamanian et al., 2021; de Soto et al., 2024), suggesting that it may be worthwhile to assess the potential role of carbonates as a source or sink of C in a context of global climate change.*

You did mention alternative thermal analysis to quantify SOC and SIC content in a single subsample, but you did not discuss your results with Rock-Eval against this alternative thermal method. Why a CHN elemental analyser (and we do not know which one) was selected for comparison and not an analyser with a thermal analysis ramp (such a LECO)? Does using Rock-Eval thermal analysis offer extra benefits over other thermal analysis that exists? Please add some words on the availability of the Rock-Eval equipment and the cost of the analysis. Is it a method easily available in several soil analysis labs?

*Answer : The aim of the draft was to compare data from an accredited reference laboratory using standardized methods. In the revised version, we have provided more details on the protocols used by the LAS to quantify SOC and SIC in the reviewed version of the technical note starting line 61. Briefly, total carbon is determined using a CHN analyzer (Thermofisher Flash 2000) and inorganic carbon is quantified using a Bernard calcimeter. In carbonated samples, organic carbon is determined as the difference between total carbon and inorganic carbon. Our mention of thermal analysis concerned previous studies using thermogravimetry or Rock-Eval®. We are not aware of any works published in the scientific literature using LECO analyzers to quantify SOC and SIC in one batch.*

I did not understand in L. 36-37, if Koorneef et al. has proposed some corrections to estimate SOC and SIC and why you did not discuss your results as you did with Hazera et al. corrections with Koorneef corrections. Did you suggest that Koorneef et al. and Hazera et al. did not present validate results in their paper? Please discuss the domain of validity of each study and how you improve the method.

*Answer : There is indeed no correction proposed in Korneef et al. One of the conclusions of this manuscript is that it would be useful if a correction can be developed. From our point of view, the correction proposed by Disnar et al., and then reused in Hazera et al., is not validated. For a method to be validated, we consider that it must be independently evaluated and that metrics informing on the adequacy between corrected values and target observations must be presented (RMSE, BIAS).*

Additional information on the LAS analysis should be add to be more convincing. We have to trust the authors about the accuracy of the LAS data, no mention on the incertitude or if there is any replicates. The method used to get SOC and SIC content are not mentioned (calculation, direct measures with or without pretreatments for SIC or SOC ?). As it is a methodological work, it should be nice to insert some reference geostandard samples in the data set to check the accuracy of the LAS and Rock-Eval® analysis.

*Answer : We agree, we have provided more details in the revised manuscript. The measurement uncertainties of LAS have been determined and are added in the revised manuscript starting line 65:*

*Inc. SOC = 0.02 \* ([SOC]+0.12\*[CaCO3]) + 0.49 + 0.12 \* (0.016 \* [CaCO3] + 0.63) in g/kg*
*Inc. SIC = 0.12 \* (0.016\*[CaCO3]+0.63) in g/kg*

*A reference sample (IFPEN_160000) is measured regularly in Rock-Eval® batch analyses for quality control. The target values are 3.28 ± 0.06 wt% and 3.26 ± 0.07 wt% for TOCre6 and MinC respectively. The machine is recalibrated when TOCre6 and MinC values are outside this narrow range meaning that the acceptable measurement error is ca. 2%. This has also been specified in the revised manuscript starting line 87.*

Are they any replicates on the LAS and Rock Eval analysis? What is the incertitude associated to each of the analysis? This is a methodological work, so readers expect high standard on the description of the methods.

*Answer : The incertitudes associated to LAS and RE measurements have been given above. They have also been added to the revised version.*

L.70 Please can you add some indication of the soil aliquots. You only say ca. 60 mg. I presume it is ground soil (200 µm). Is it OK for any soils, even the soil with very high SIC and very low SOC content? Or the inverse? What are the limit of C detection ? limit of C content of each C pools to be properly measured. Is it possible to have a saturation of the signal for one of the C pool and in a same time to be at the lowest detection limit for the other C pool? As your paper is a technical note, this consideration are worth to be add for the future users of your method.

*Answer : The Flame Ionization Detector (for HC) can be saturated. That is why for pure organic samples, only 10 mg are used. The other detectors get saturated at very high C content that are not reached when working with 60 mg of sample. From our experience, we do not feel comfortable with samples with TOC lower than 1.5 gC/kg. In the range of SOC (5.37 to 44.64 gC/kg) and SIC (0.5 to 96.8 gC/kg) contents used in our study, we did not have any issues. We agree that we should present very clearly the range for which our correction has been validated. However, we consider that the investigated SIC and SOC ranges are quite large making our correction valid for the vast majority of agricultural topsoils. We have specified the ranges of SIC and SOC in the caption of Figure 1..*

L.96-97 How many values of SIC inferior to zero were in the data set? SIC could be calculated inferior to zero for what range of reference SIC values?

*Answer : Thanks for your remark. Before correction, SIC can't be inferior to zero. Even in soil without SIC, the MinC parameter would not be zero because a tiny part of OC is erroneously attributed to MinC when using Rock-Eval. This is a well known artifact. This amount of C erroneously attributed to MinC comes from the S3 signals. If we do not have a contribution to the S5 signal to the MinC it means that the soil does not contain SIC. We checked that it was true for the 48 samples that got negative SIC values after corrections. As it was the case, it means that in our sample set, only samples with no carbonates can get (very small) negative SIC values after correction. It is therefore not a problem to correct them to 0.*

L.98 « all the RE parameters » and L. 106 "after a correlation analysis…." Please specify the RE6 parameters tested. Furthermore if the reader is not familiar with Rock Eval analysis, he/she cannot understand what you mean without a minimum of explanation on these parameters.

*Answer : We have provided some examples of RE parameters (for instance Hydrogen and Oxygen Index) in line 118 of the revised manuscript, and specified the meaning of the sentence on line 106-107 (now beginning line 129). We tried multivariate models and observed that they were not better than univariate models, we have provided more details in the same paragraph.*

L.105 How have you proceeded to choose the calibration and the validation sample sets?

*The calibration and validation sample sets are chosen fully randomly by using 160 samples during the training phase, and using the remaining 80 samples for validation. We check that the calibration and the validation data sets follow similar distributions of SIC and SOC. This*

*is shown in the Notebook, note that the calibration and validation sets correspond to the train and test sets.*

L.114 are you sure that the conditions given in the application of the correction suggested by Hazera are OK for all of the soil samples of your data set (soils enriched in poorly degraded organic compounds)? Please specify which the land uses are represented in your data set and why the soils are probably enriched in poorly degraded organic compounds.

*Answer : We specified (line 55 in revised paper) that we used topsoil samples from agricultural soils. The samples used in Hazera et al. are also coming from agricultural soils and fall within a similar SOC range. For this reason, we used the same correction as they did. However, in the revised manuscript and the Notebook we have included both corrections and conclude that the ML approaches still perform the best.*

Some clarification and homogenisation should be done between the term and the abbreviation using CHN, LAS, Elemental analyser. For example L. 145 "reference CHN value", but in the figures LAS is mentioned. The same for TOC-RE6 and SOC-RE6. Please try to be clear with all the abbreviation to help the reader to follow.

*Answer : We agree, thank you for this remark. This has been corrected in the revised version.*

L.135 The sign "sum" is missing. Please correct the Equation.

*Answer : Thank you, we have corrected it.*

L.151 I did not understand if you realised three different models, with three different corrections (non carbonated, carbonated, all soils). I did not see the discussion on the different corrections. One of my problem with your work is that these corrections were not explicit. They will be included in the Geoworks software OK but we do not see how it is calculated. The machine learning procedure is clear but the statistical corrections were somehow cloudy because not explicit in the text. I am not sure that your work could be reproducible.

*Answer : Some machine learning models cannot be easily presented in an analytical form. However, our work is fully reproducible as all the data and the codes are available.*

Why figure 2, 4 and 5 have different scales? (100 g kg-1 and 50 gkg-1 for SIC and SOC respectively in Fig.2 and 70 and 35 in Figure 4). Please be consistent.

*Answer : Figures 2 and 5 contain the complete dataset, whereas figure 4 only the 80 samples used in the testing set. Since there is only a single sample with SIC > 90 g/kg and two samples with SIC > 40 g/kg, it is highly likely that these samples do not get selected in the testing set. .*

The predicted corrections of the ML model were not explicit. Is it possible to do so? Why not?

*We thank the reviewer for this question, we shall modify the revised version to make these notions clearer. While it is theoretically possible to represent a Support Vector Regression (SVR) in its analytical form, doing so would require a function whose size depends on the size of the learning set. In our case, this would mean an analytical function containing 160 expressions for each of the points in the learning set.*

*As a workaround for this issue, the accompanying Notebook contains the Partial Dependence Plots (PDPs) for the SVR model, as well as the other four models we tested. The PDPs show the predicted target as a function of a feature of interest, in our case the SIC and the SOC. It also allows us to visualize the difference between the four proposed ML models.*

L.160 "test data" what do you mean? Data in Hazera et al. were compared with reference data, but the data were not statistically adjusted as you did, the corrections proposed were defined from literature (e.g. Disnar et al. 2003).

*Answer : The sentence is not appropriate and has been corrected in the revised version in line 202. In fact, the correction proposed in Hazera et al. (originally Disnar et al.) was not learned on one dataset and evaluated on another. To this respect, it is logical that it is not optimal.*

Please give a domain of validity of the method. Why do you say is it the first method to be validated? The corrective approach of Hazera did not work as well as yours on your data set, but it seems OK for their own data set. Please discuss and give clue to explain what could explain this discrepancy. Is it a question of SOC and SIC range, of type of soil, or of land use or land cover, type of SOC, type of SIC?

*Answer : There is a conceptual difference between the approach proposed in Hazera et al (originally Disnar et al.). and our approach. We produced a correction method with a proper evaluation on an independent dataset and evaluated the transferability of our validated correction method. This was not done in Hazera et al. Our study shows that the correction of Hazera et al. improves the initial data, however it is not optimal. At first sight, one might even consider that the Hazera correction is OK for our dataset. However, we consider that it is more relevant to implement the best correction method and our SVM correction is clearly better.*

Please discuss more your results, e.g. the incertitude associated to the values to the reference data and to the RockEval data. What are the limits and the perspectives of your work. Could these corrections be applied also on soil fractions? On any kind of soil with different organic matter and different calcareous minerals? Are 160 French carbonated soils enough? Does it need further calibration, probably yes on the soil fractions as it is tested in Koorneef et al. What are the possible perspective of this work? You also could further understand why some samples are better predicted than others.

*Answer : We have added a few lines of discussion regarding the uncertainties. The uncertainties associated with our method are presented in the figures. Those associated with the LAS and RE measurements are specified starting line 66 and 87, respectively. There is no reason to consider that our correction cannot be applied to fractions as long as SIC and SOC contents are in the range of our calibration set. We have detailed these ranges in the caption of Figure 1.*

*Regarding the size of the data set, we have tested the precision of the proposed correction when the data set size ranges from 16 samples to 160 samples. We conclude that having at least 100 samples is ideal, while already 50 soil samples provide reliable SVM corrections. The benefit of 160 soil samples is then the fact of representing as many different soil types as we had available. We have added new figures in the supplementary materials of the accompanying Notebook to address these questions.*

To conclude, I think this work is worth to be published after major revisions. It gives promising result and improvement to measure SOC and SIC in single aliquot for a large range of soils. I regret that the corrections were not explicit (a statistical adjustment but the coefficient were not given and discussed against the coefficients existing in the literature for SOC or SOC and SIC content). It works on 260 French soil samples. The Rock Eval must be corrected by statistics… is it not possible to fix the cycle of analysis to avoid these corrections? I also regret that the discussion on the proposed method is poor and no

perspectives of the work are proposed. It looks like that because the statistical adjustment is OK for 260 French soil samples, the problem of measuring SIC and SOC content in soil with Rock Eval is no more a problem for any kind of soil materials. Please discuss more the result to avoid the feeling that the paper is an advertiser for RockEval and Geoworks. This very interesting work is really promising despite my many comments, it deserves further discussion.

*Answer : Thank you for your feedback that will allow us to improve our draft. We have tried to reply to your feedback by modifying several sections of the technical note and adding the information that was missing. We have also updated the accompanying Notebook to include more information on several of the remarks proposed, and we hope that these modifications answer your questions and improve the quality of our publication.*

**Reviewer 2**

The results presented by Stojanova et al. (2024) constitute a significant contribution to the use of Rock-Eval thermal analysis in soil science, as they provide the first comparative study of correction methods for organic carbon (Corg) and inorganic carbon (Cinorg) contents measured by RE in soil samples.

This question is highly relevant because, from the first RE applications to soil samples, Disnar et al. (2003) already observed a significant discrepancy between organic carbon (Corg) contents measured by RE (TOC parameter) and by elemental analysis (LECO). These authors then proposed an "empirical correction" based on simple linear regressions between RE and LECO measurements using a dataset (n = 100) representative of the main types of horizons (organic, organo-mineral, and mineral) sampled under contrasting pedoclimatic conditions.

The manuscript presented by Stojanova et al. (2024) certainly addresses the shortcomings of this initial approach. Firstly, the authors clearly formalize their objectives to minimize discrepancies between organic carbon (Corg) and inorganic carbon (Cinorg) contents by using elemental measurements as a reference. Secondly, the study is conducted on a large panel of samples (n = 240) covering a wide range of Corg (0-50 g.kg-1) and Cinorg (0-80 g.kg-1). Thirdly, the study involves comparing the performances of several models using objective statistical criteria. Results show that these performances, analyzed for different soil categories, allow the identification of a significantly more efficient model than others, thus providing a simple and effective post-analysis "statistical correction" for the studied soil types.

Reading the manuscript, however, raises some incidental questions that can be shared with the authors to strengthen this technical note.

*Answer : thanks for your positive feedbacks and interesting comments.*

Among the correction procedures tested by Stojanova et al. (2024), four are proposed by the authors, while the fifth is presented as a "correction model" proposed by Hazera et al. (2023), which constitutes a shortcut and raises an attribution issue. Indeed, the work presented by Hazera et al. (2023) focuses exclusively on adjusting the Rock-Eval analytical protocol to improve the accuracy of the initial measurement. The question of post-analysis corrections is addressed as a technical contingency based on the literature. In the "Materials & Methods" section, Hazera et al. (2023) present the empirical correction protocol proposed by Disnar et al. (2003), and then another protocol of "parametric correction" based on a prior interpretation of the RE data (Sebag et al., 2022). Hazera et al. (2023) explicitly state that they use the empirical parameters proposed by Disnar et al. (2003) to correct RE measurements. Therefore, by adopting the protocol used by Hazera et al. (2023), Stojanova et al. (2024) compare the performances of their statistical models to the empirical procedure proposed by Disnar et al. (2003). It seems important to correct this attribution error to avoid any confusion regarding the origin of the correction method.

*Answer :The correction proposed in Hazera et al. indeed comes from Disnar et al. and SOTHIS. We have made it clearer in the revised version starting line 141. Moreover, our work also confirms that all carbonates have evolved as CO2 using the classical "SOIL" mode of the Rock-Eval(r) (described in detail in Cécillon et al., 2018) and that the adjustment of the usual analytical protocol is not necessary. We have also added a sentence beginning line 217 to address this issue.*

By using the formulas presented by Hazera et al. (2023), Stojanova et al. (2024) implicitly employ the empirical correction procedure proposed by Disnar et al. (2003) without explicitly stating it. However, this procedure explicitly comprises two distinct and successive steps: the first applies unconditionally to all samples, while the second applies only under certain conditions to specific samples after a prior examination of qualitative RE parameters. In the present form of the manuscript, it appears that Stojanova et al. (2024) systematically applied the second step to all samples without prior verification of the conditions for its application. It is crucial that the procedure proposed by Disnar et al. (2003) is implemented in accordance with its technical recommendations. It is highly likely that the results will not be radically different from those currently presented, but it will minimize any uncertainty when comparing the models' performances.

*Answer : There is indeed a different correction to apply depending if one considers that the soil samples contain poorly degraded organic compounds or not. As we worked on topsoil samples only, we consider that our samples contain poorly degraded organic compounds, as in Hazera et al. We have since checked both corrections (with or without considering that soil samples contain poorly degraded organic compounds) and accordingly modified the revised version. Nevertheless, we consider that if the use of a correction method depends on something that is difficult for the user to determine, this means that such a method is unsuitable. We invite the reviewer to consult the revised version, notably the sections starting lines 141 and 155.*

To apply the Disnar's correction extended to Cinorg, Stojanova et al. (2024) propose using a threshold value of 2 gC/kg of SIC to determine which samples are calcareous or non-calcareous. However, the MinC parameter includes a portion of released Corg, particularly during the pyrolysis phase (Hazera et al., 2023; Koorneef et al., 2023). So, does the use of MinC to distinguish calcareous from non-calcareous soils introduce uncertainty in samples with high TOC? Since the comparison of models is conducted for three soil populations (non-calcareous, calcareous, all), have the authors verified the accuracy of these categories through mineralogical analyses (such as XRD) or through a detailed examination of thermograms (as in Pilot et al., 2014)? Another question concerns the minerals. Do the proposed models integrate soils containing minerals other than calcite? Thermograms of dolomite or siderite are quite distinct (Pilot et al., 2014). Could this impact the performance of the machine learning models?

*Answer : We thank the reviewer for this comment. We estimated that this would not make a significant difference, but there are indeed more appropriate ways of determining whether a soil sample is carbonated or not than the threshold we have chosen. The amount of C erroneously attributed to MinC comes mostly from the S3 and S3' signals (PyroMinC). We have determined if the sample is carbonated by comparing the OxiMinC and the PyroMinC. When OxiMinC is < 0.1 wt%, the soil sample can be regarded as non-carbonated whatever the PyroMinC values, which is organic carbon in that case. In the accompanying Notebook, we have tested both approaches and we find that there is no statistically significant difference in using one threshold or the other.*

*From the thermograms, we determine that we don't have neither dolomite nor siderite. We have specified this in the revised version in line 92.*

One of the strengths of the work by Stojanova et al. (2024) lies in their explicit and unambiguous statement of the objectives of their approach: to identify the correction method that minimizes the discrepancies between the corrected RE measurements and the standard

measurements used as reference. By correcting the RE measurements in this manner, the authors achieve a very satisfactory approximation to standardized measurements (ISO). This significant advancement will facilitate the practical use of RE data while awaiting their possible standardization. Concerning the comparison of models, the analysis is based on their respective performance in minimizing discrepancies with the reference method. However, the results are not analyzed relative to each other. Would it be possible to verify that the differences between the methods are statistically significant considering the analytical precision of the methods used?

*Answer : Pairwise comparisons show that the differences between SOC and SIC values provided by the different methods are most often highly significant (p<0.001). In particular, outputs of the correction methods presented in Hazera et al. tend to overestimate SOC and are highly significantly different from those provided by the other correction methods. We have added some figures and paired T-test results on this aspect in the Notebook.*

However, minimizing the discrepancies between the corrected RE measurements and the standard measurements raises a more fundamental question: it is well-known that standard protocols for measuring SOC and SIC entail several inevitable errors related, on one hand, to sample pretreatment (removal of Corg or Cinorg), and on the other hand, to measurements on two different aliquots. Therefore, in seeking to minimize the discrepancies between RE measurements and the standard method, the authors import errors associated with the latter for calcareous soils. From a methodological perspective, it would be judicious to indicate this limitation in the technical note. Indeed, this inherent limitation to the stated objectives highlights the need for further studies to improve the initial measurement and correct the systematic misattribution of Corg as Cinorg.

*Answer : We agree with the reviewer on this point and have included a few more details in the revised version starting in line 61. This is also the reason why we decided to have SOC and SIC measured in the French soil analysis laboratory. We are not sure that the "initial measurements" can be substantially improved. We have included the uncertainties linked to the SIC and SOC measurements provided by the LAS (line 65), as well as details on the reproducibility of the RE procedure and the usage of a reference sample in all RE batch analyses (line 87).*

This is why the end of the abstract raises perplexity when the authors write: "that the proposed correction significantly increases the accuracy of the Rock-Eval method on the initial dataset, and that it can be successfully applied to data originating from different Rock-Eval machines, without changing the routine analytical protocol." This statement seems to be in contradiction with the objectives presented. The correction did not increase the accuracy of the RE method; it reduced the discrepancies of the measurements with a reference method that has its own errors. The RE method would increase its accuracy if the analytical protocol or calculation methods avoided confusion regarding the forms of carbon.

*Answer : In our view, the LAS provides the reference measurements, and so being able to supply SOC and SIC from Rock-Eval® data as close as possible to the LAS values corresponds to an increase in the accuracy of Rock-Eval measurements. However, we understand the subtlety of the reviewer's remark and have modified the sentence accordingly as shown in line 8 of the reviewed version.*

In conclusion, one may question the format chosen by the authors to publish their results. Does the Technical Note format allow for the development of all the necessary discussions to truly highlight the results? This format, which minimizes the scientific issue in favor of the results, accentuates at the same time the "advertisement for the Rock-Eval device and

Geoworks software" commercialized by the company that funded the study, both of which are used for commercial purposes by another party of the study's co-authors (which is not indicated in the section dedicated to potential conflicts of interest).

*Answer : We thank the reviewer for this remark, however we are not sure to fully understand what they mean by "the necessary discussions to truly highlight the results". We think we have given a clear answer to a specific technical question that could greatly facilitate research in carbonated soils. Given the nature of the work itself, we think the Technical Note format is fully appropriate and encompasses the entirety of our contribution. The problem of carbon quantification in carbonated soils is a major one that severely limits the community's ability to study these soils. It is our shared belief that the work we present is far from an advertisement for a machine and a software, and instead it represents a much-awaited solution for the scientific community. Moreover, we have made sure to make the data and routines freely available in an open-source language and platform, so any user can apply the proposed corrections without having to buy Geoworks.*

---

## Author Response (AR2)

**Authors' comments and replies to Reviewer 1**

*I thank the authors for their responses to the reviewers and the revisions made to the manuscript. Overall, these revisions enhance the presentation of the results. As I noted initially, these results will enable the consideration of RE data while the method is being standardized.*

*However, two points still raise questions for me.*

*First, regarding the differences between correction methods: Are the discrepancies between corrected LAS and RE, and among the different corrections, statistically significant if they fall within the combined margins of error of both methods?*

We have included paired T-tests comparing the Disnar corrections and the SVM correction that we propose. These clearly demonstrate the difference between the two corrections. When it comes to the margin of error of the two carbon-estimation methods, the proposed corrections, both Disnar's and ours, aim at removing a general bias that is widely known to exist. It is the authors' opinion that the sample-wise corrections might sometimes fall within the margins of error, however the systematic bias that exists in the population of samples still needs to be corrected.

*Second, the final sentences of paragraph 3.2 seem disconnected from the results presented. The conclusion of a technical note may not be the best place to categorically assert that post-acquisition statistical correction is preferable to modifying the analytical protocol to improve measurement.*

We thank the reviewer for this comment. We have rephrased this statement and placed it in the Methods section so as to be more coherent with the format of the publication.